# Research on Seawater Intrusion Suppression Scheme of Minjiang River Estuary

**DOI:** 10.3390/ijerph20065211

**Published:** 2023-03-16

**Authors:** Ziyuan Wang, Yiqing Guan, Danrong Zhang, Alain Niyongabo, Haowen Ming, Zhiming Yu, Yihui Huang

**Affiliations:** 1College of Hydrology and Water Resources, Hohai University, Nanjing 210003, China; 2Minjiang River Estuary Hydrology Experiment Station, Fujian Hydrology and Water Resources Survey Bureau, Fuzhou 350011, China

**Keywords:** estuary, seawater intrusion, chlorine level, critical river discharge, random forest

## Abstract

Seawater intrusion in the Minjiang River estuary has gravely endangered the water security of the surrounding area in recent years. Previous studies mainly focused on exploring the mechanism of intrusion, but failed to provide a scheme for suppressing seawater intrusion. The three most relevant determinants to chlorine level, which represented the strength of seawater intrusion, were determined using Pearson correlation analysis as being the daily average discharge, daily maximum tidal range, and daily minimum tidal level. Considering the lower requirement of sample data and the ability to handle high-dimensional data, the random forest algorithm was used to construct a seawater intrusion suppression model and was combined with a genetic algorithm. The critical river discharge for suppressing estuary seawater intrusion determined using this model. The critical river discharge was found to gradually increase with the maximum tidal range, which in three different tide scenarios was 487 m^3^/s, 493 m^3^/s, and 531 m^3^/s. The practicable seawater intrusion suppression scheme was built up with three phases to make it easier to regulate upstream reservoirs. In the scheme, the initial reading of river discharge was 490 m^3^/s, and it rose to 650 m^3^/s over six days, from four days before the high tide’s arrival to two days following it, and before falling down to 490 m^3^/s at the end. Verified with the 16 seawater intrusion events in the five dry years, this scheme could eliminate 75% of the seawater intrusion risk and effectively reduce the chlorine level for the remaining 25% of events.

## 1. Introduction

Coastal estuaries are prone to seawater intrusion [1]. With the rising tide, seawater with a high chlorine content flowed upstream and mixed with freshwater in the river, turning the upstream water salty and unusable for drinking or irrigation [2]. This threatened the freshwater demand on coasts, especially in some water-scarce areas. The issue of seawater intrusion has gradually spread across the globe in recent years due to increased human activities, the influence of sea level rise, and climate change [3]. Seawater intrusions were reported in coastal aquifers along both the east and west coasts of the United States [4,5]. Intrusions occurred in Mexico, Australia, and Libya [6,7,8]. Additionally, in several regions of Europe and Asia, seawater intrusion has occurred and may be a threat in the future [9,10,11,12]. Countries provided early warning of threatening seawater intrusion, of which Chinese water quality norms state that water cannot be used for manufacturing or as a source of drinking if the chlorine level is greater than 250 mg/L [13].

Many studies analyzed the causes of seawater intrusion in order to find a solution for it [14]. Xue et al. investigated the physical mechanism of seawater intrusion in the Yangtze River estuary using a finite-volume coastal ocean model (FVCOM) [15]. The results show that a combination of upstream flow, tidal currents, mixing of seawater and freshwater, wind, and salt distribution in the inner shelf of the sea affected the seawater intrusion. Xu et al. assumed that different tides and runoff intensities would impact the mixing type of seawater and freshwater, and built an MIKE21 model to simulate runoff, tide, and wind on the activity and concentration of the seawater [16]. Yin et al. created a three-dimensional numerical simulation experiment to investigate the combined effect of water flow and tide on seawater intrusion in a runoff-dominated stratified estuary. They then used this model to explore the best timing for increasing river discharge for suppressing seawater intrusion [17]. Wang et al. considered the cumulative effect of the above factors on chlorine levels and developed a more realistic chlorine level model in the Pearl River estuary [18].

Numerous studies have shown that, to a certain extent, high river discharge could weaken seawater intrusion [19,20,21]. Meanwhile, the concept of critical river discharge to suppress seawater intrusion, the minimal river flow necessary to ensure that chlorine level in the mixed river water does not exceed the threshold, has been gradually put forth [22,23]. Aesawi et al. used MIKE21 to simulate the hydrodynamics and seawater intrusion in the Saudi Arabian estuary and used the results to calculate the minimal freshwater discharge required for seawater suppression [24]. A two-dimensional numerical model has been used by Liu et al. to investigate the relationship between river discharge and the distance of seawater intrusion [25]. These studies focused mainly on the suppression effect of flow on tides, but could not give solutions to the water scarcity situation.

It was important to divide the models discussed above into two groups: statistical and numerical. Numerical models, which were often two- or three-dimensional models based on water outflow, tide, meteorology, topography, and other multidimensional information, were used to evaluate the effects of one or more elements in the mechanism of seawater intrusion [26,27,28]. The statistical models, which were mostly used for the study of the seawater suppression strategy or the prediction of seawater intrusion, had been based on actual data, taking into account the statistical correlation among variables [29]. Numerical modeling required a large amount of basic data, which was difficult to use in areas where data were lacking. With the rise of computer technology, machine learning has been employed for multivariate analysis of nutrient distribution in water environments across temporal and spatial scales in recent decades, and the number of such studies has grown exponentially in recent years [30]. Compared with traditional algorithms, machine learning can effectively reduce the computational dimension while maintaining prediction accuracy and high stability [31]. Currently, machine learning has been widely used for numerical simulation, sensitivity analysis, pattern classification, prediction, and other studies of seawater intrusion, as well as water resources optimization management research under seawater intrusion conditions [32,33].

In the above studies, the most popular machine learning methods are artificial neural networks (ANNs), support vector machines (SVMs), random forest algorithms, etc. Compared with ANNs and SVMs, the random forest algorithm had a lower requirement for data sample capacity [34]. This was because the random forest reduced overfitting by integrating multiple decision trees, with each decision tree only using a portion of the dataset for training, making it effective in handling datasets with relatively few samples [35]. In contrast, algorithms such as ANNs required a large amount of data samples for training to achieve the same level of generalization. Moreover, because it could handle collinearity between variables, the random forest algorithm did not overfit when working with high-dimensional data.

The Minjiang River estuary is near Fuzhou, the capital city of Fujian province, China. Additionally, the most crucial source of water for Fuzhou city is the Minjiang River. The amount of sand from upstream has been reduced, river scouring has gotten worse, and riverbed undercutting has become more serious in recent years as a result of the building of upstream reservoirs and downstream sand mining activities [36]. Consequently, the estuary’s original hydrodynamic properties have been changed, the tide-affected area constantly moved upstream, and the chlorine level frequently exceeded the threshold of norms, posing major threats to the local water supply. Currently, research on the seawater intrusion into the Minjiang River mainly focuses on exploring this mechanism, with little research on measures to suppress it [37]. Therefore, studying a scheme to suppress seawater intrusion and mitigate its impact on the municipal water supply in Fuzhou is of practical significance. In this study, a statistical model of seawater intrusion suppression was developed with measured data of river discharge, tide, and chlorine level. Considering the limited sample size and high dimensionality of the data, we decided to use a random forest algorithm to drive the model. The goal was to explore the suppression scheme while ensuring downstream water use and reservoir power generation.

## 2. Materials and Methods

### 2.1. Study Area

Minjiang River, the largest river in Fujian province, originates from the Wuyi mountains near the border of Fujian province and Jiangxi province. The downstream flows past Fuzhou city and into the sea (Figure 1). Several waterworks are located in the estuary area of the Minjiang River, including Changle, Chengmen, and Mawei. The upstream Shuikou reservoir, which was built in 1993 with a catchment area of 52,438 km^2^, controls the downstream flow and is used for power generation, flood control, and irrigation. The annual flow of the Minjiang River can reach a maximum of 1733 m^3^/s, while the annual discharge is around 55 billion m^3^. However, the annual flow is extremely uneven during the year, which is abundant in June, accounting for 56% of the total runoff of the year, while in winter accounting for less than 12%.

### 2.2. Data Preparation

The salty tide frequently affects the waterworks in the downstream of the Minjiang River in dry years, causing chlorine levels to exceed the threshold. In this study, data from five dry years with frequent seawater intrusion, namely 2009, 2011, 2013, and 2017/11 to 2018/10, were selected as the data periods, which helped to explore the features of seawater intrusion. For each month in five dry years, Figure 2 shows the average number of days that the chlorine level threshold was exceeded. The dry season of Minjiang watershed usually lasted from October to February. It was evident that the cases of exceeding the threshold most frequently occurred during the dry season. The highest average days, 9 days in November and 6 days in February, were during the dry season. The number of days that exceed the threshold have progressively grown in recent years.

From Figure 3, it can be seen that fluctuations in discharge flow and tide downstream of the Minjiang River are the main factors affecting seawater intrusion (chlorine level), which are used to seek the change in chlorine level in this study. This plan can be supported by the conclusions of Chen et al. [38] by finding that the water flow, which suppressed saltwater, constrained the tide’s gradient force, which promoted the infiltration of seawater. Therefore, flow data, tidal data, and chlorinity data were obtained from Zhuqi hydrological station, Wenshanli tidal station, and Changle waterworks, respectively, as shown in Figure 1.

### 2.3. Factors Selection and Data Sources

In this study, the daily average discharge data served as a proxy for the river discharge, and the tidal dynamics are represented by the daily average tidal level, daily maximum tidal range, and daily minimum and maximum tidal levels, since tidal level and tidal range were the two main elements used to describe tidal dynamics. The Pearson correlation analysis has been used to determine the correlation between the chlorine level data and each of the above five influencing factors. The computation was implemented with the following equation:(1)$$r=\frac{\sum_{i=1}^{n}\left(X_{i}=\bar{X}\right)(Y_{i}=\bar{Y})}{\sqrt{\sum_{i=1}^{n}{\left(X_{i}=\bar{X}\right)}^{2}}\sqrt{\sum_{i=1}^{n}{\left(Y_{i}=\bar{Y}\right)}^{2}}}$$

The Pearson correlation coefficient is a statistic that measures the linear relationship between two variables. It varies from −1 to 1, where −1 means a perfect negative correlation, 1 means a perfect positive correlation, and 0 means no correlation.

This study considered the cumulative effect of flow and tidal activities, with both the flow and tide of the present day and previous days influencing the present-day chlorine levels [39,40]. The results of the Pearson correlation analysis conducted in this study to ascertain the relationship between the chlorine level of the present day, the influence factors of the previous days, and the present day are shown in Figure 4. This study included 1102 sets of samples. The chlorine level was significantly correlated with influence factors from the present day to the previous five days, indicating that flow and tide could have a considerable impact on the chlorine level of the present day, going back as far as five days.

The influence of tidal range was represented with the average and the maximum tidal ranges, two types of influence factors that had a positive correlation with chlorine level. In Figure 4, the correlation coefficients of the maximum tidal range decreased with time, but the minimum value was 0.177, which was still greater than the maximum value of 0.122 in average tidal range correlation coefficient. Additionally, all the correlations in the maximum tidal range data were significant at 0.01 levels, which indicated that all data in the maximum tidal range were closely related to chlorine level changes, while only the correlations on the present day were significant at 0.01 levels in the average tidal range. Therefore, the maximum tidal range was chosen as the only tidal range data that was finally included in the model in order to prevent data redundancy from decreasing the model’s operational efficiency. In the same situation, the model’s tidal level data have been chosen to be the minimum tidal level. Although the correlation coefficients of the maximum and minimum tidal levels were both significant at 0.01 levels, Figure 4 showed us that the correlation of the minimum tidal level was higher than that of the maximum tidal level on each day. Although the correlation value of the minimum tidal level was negative, this only represented that it was opposite to the trend of chlorine level change. The absolute values of the correlation coefficients of the daily average discharge were all greater than 0.25 and significant at 0.01 levels, which indicated that during the study period, runoffs had continuous and close impacts on chlorine level, so these data were also added to the model.

The calculations showed that adding or removing the tidal range and level data from five days ago to the model had essentially no impact on the model’s accuracy. As shown in Figure 4, their correlation coefficients were significantly lower than on the other days. Therefore, to further improve the model’s operational efficiency, data on the tidal range and level that occurred five days ago were eliminated in this model.

Finally, 16 sub-factors connected to the chlorine level have been chosen to build the model, including data on daily average discharge from the present day to the previous five days, maximum tidal range, and minimum tidal level data from the present day to the prior four days.

### 2.4. Seawater Intrusion Suppression Model

#### 2.4.1. Random Forest Algorithm

A random forest algorithm (RF) was used to construct the chlorine level statistical model in this study. It was first introduced by Breiman in 2001 as an integrated machine learning algorithm based on decision trees [34]. As Figure 5 shows, this approach comprised a number of built-in decision trees, each of which was produced using random sampling and independent random characteristics and whose results were statistically averaged to produce the result. One benefit of the random forest was its capacity to locate, examine, and choose which components in the sample should have higher weights. It did not require a big sample size and could maintain high accuracy with a small amount of known information.

#### 2.4.2. Chlorine Level Statistical Model

A total of 1102 sets of daily data from 2009 to 2018 were selected, and they were randomly divided into training and test datasets in proportions of 7:3, or 772 sets and 330 sets, respectively. Data were normalized to the range of [0, 1] while taking into account the unit differences between the factors prior to being entered into the model. The output results must be back-normalized after being calculated. The following hyper parameters had been selected as the model was created by Python. The decision tree count was set to 125 following trial computation. Max depth was set to eight, min samples split to five, OOB score was set to switch on validation, which can boost the applicability of the model, and the remaining hyper parameters were set to their default values. The results of the model’s simulation of the training and test datasets have been shown in Figure 6. The decision coefficient R^2^ has been established as follows and used to evaluate the precision of the model fit.
(2)$$R^{2}=\frac{SSR}{SST}=\frac{{\sum \left(y_{pre}-\bar{y}\right)}^{2}}{{\sum \left(y-\bar{y}\right)}^{2}}$$

As shown in Figure 6, the values of R^2^ for the model’s training and test datasets were 0.86 and 0.74, respectively, which were commendable outcomes. The measured and computed findings showed that the model could mimic chlorine level change in response to river discharge and tidal fluctuations since they were in reasonably close agreement. The model’s numerical values for some samples with high chlorine level values were lower than they ought to have been since there were not many samples in the training set with high chlorine levels. Due to the small percentage of high values, in general, this random forest model could help to simulate seawater intrusion and was applicable.

#### 2.4.3. Seawater Intrusion Suppression Model

As shown in Figure 1, the Shuikou reservoir upstream could control the Minjiang River’s downstream daily river discharge, which provided the possibility of using the river discharge to suppress seawater intrusion. Due to the fact that the result of the seawater intrusion suppression model was a set of many daily average river discharge values with limitations, this study used a genetic algorithm to solve it. The genetic algorithm replicated the genetic evolution of selection and variation to find the best solution through continuous iteration. It was a stochastic search strategy for limited optimization problems [41]. In this model, a genetic algorithm was used to generate a set of random daily average discharge value sets. The target result was set to minimize the total runoff of sets with the following function:(3)$$target=Q_{min}=\sum_{i=1}^{n}q_{i}$$
where Q_min_ is the total minimum runoff (m^3^/s), which contains the daily average discharge value of i days (m^3^/s).

The chlorine level statistical model was coupled to the genetic algorithm as part of the model, receiving value sets from the genetic algorithm and calculating the chlorine level data. Then, the chlorine level data were passed back to the genetic algorithm as a constraint which was used to sieve out the unqualified sets. The constraint function was as follows:(4)$$0\leq c_{i}\leq 250\;mg/L$$
where c_i_ is the day’s i of chlorine level (mg/L).

In this study, the model was planned to be used in the simulation of three tide scenarios. In the first scenario, the average tidal range and level for each month in five dry years were calculated. Additionally, the actual measured mechanisms of seawater intrusion during the dry years were the second and third. The tidal cycle lasted for 15 days, and the initial day of the cycle required the flow data from the preceding five days because the last day of the cycle would have had an effect on the chlorine level measurements four days later. Therefore, it had intended to employ a 24-day computation period.

## 3. Results and Discussion

### 3.1. Effect of River Discharge

A chlorine level statistical model was used to model the chlorine level at the Changle waterworks while the reservoir discharged water at different flow rates. The tidal range and tidal level values in the model were derived from the measured data. This clarified how water discharge affected seawater intrusion. The maximum tidal range was utilized as the horizontal axis in Figure 7 to indicate the tidal activity. Both the maximum tidal range and the minimum tidal level may equally depict the tidal activity, and the maximum tidal range was more generally used to represent tidal activity in related studies. The biggest tidal value in the five days was selected as the horizontal coordinate value because it could better portray the intensity of tidal activity, as five days of data on tidal range would have needed one day of chlorine level in the model. The value of river discharge ran vertically along the axis. In this section, we assumed that the flow rate of reservoir discharge was constant in each single simulation.

The chlorine level at the Changle station was typically less than 200 mg/L, which did not pose a hazard to the waterworks when river discharge was above 600 m^3^/s and the tidal range was lower than 4.6 m. The chlorine level would be greater than 350 mg/L and the highest would exceed 800 mg/L when the river discharge was less than 560 m^3^/s and the tidal range was higher than 4.6 m. This indicated that a large flow could effectively suppress seawater intrusion.

As the tidal range rose, it was evident that the critical river discharge to meet the threshold gradually increased as well, according to the national threshold of the critical chlorine level, which was 250 mg/L. The crucial flow rate increased with a tendency of first slowing down and then speeding up. The critical river discharge ranged from 440 m^3^/s to 520 m^3^/s when the tidal range was between 4.0 and 4.3 m, and the river discharge increased by about 26.7 m^3^/s for every 0.1 m of tidal range rise during this range. The river discharge was between 520 m^3^/s and 550 m^3^/s when the tidal range was between 4.3 and 4.6 m. During this time, the increase in the critical river discharge brought on by the shift in tidal range was milder. It increased by roughly 10.0 m^3^/s for every 0.1 m of increased tidal range. The critical flow ranged between 550 m^3^/s and 650 m^3^/s when the tidal range was between 4.6 m and 4.9 m. Each 0.1 m increase in tidal range resulted in an increase in critical flow of around 33.3 m^3^/s. The critical flow ranged between 650 m^3^/s and 720 m^3^/s when the tidal range was between 4.9 m and 5.0 m.

### 3.2. Minimum River Discharge for Seawater Suppression

The lower Minjiang River discharged insufficiently during the dry season, and the chlorine level of the estuary was prone to exceeding the threshold during high tides due to the severely unbalanced intra-annual distribution of precipitation in the Minjiang River basin.

It is vital to investigate and optimize the minimal discharge for seawater intrusion suppression scheme during the dry season since, in the event of a high tide, the available water in the reservoir will not be sufficient to support the requirement for a constant flow in the previous subsection. Based on the seawater intrusion suppression model, this scheme dynamically adjusted the daily discharge on a daily basis to make sure that the chlorine level was always below 250 mg/L by analyzing the tide’s movement.

Starting with a river discharge of 308 m^3^/s, the minimum river discharge necessary for navigation in the Minjiang River’s downstream, Figure 8a depicts the first tide scenario which was a hypothetical tide, generalized from the actual tide information and representing the average of the actual tides. Figure 8b shows the optimized seawater intrusion suppression scheme and the corresponding chlorine level. The river discharge rose along with the increase in tidal range. Before the high tide, the river discharge had increased to a maximum of 671 m^3^/s, and the chlorine level was always kept below 250 mg/L. In an effort to save water resources as the tidal range shrunk, the river discharge steadily dropped and eventually returned to its initial level. The entire procedure took 24 days, and the daily average discharge was 487 m^3^/s. The whole process complied with the minimum discharge criteria for seawater suppression.

Figure 9a depicted the second tide scenario, which was an actual tide that took place in 2009. Since the tidal range reached 5 m at high tide, the chlorine level was rapidly elevated following high tide, peaking at 492 mg/L. (Figure 9b). Figure 9c depicts the optimized scheme used in this instance, which progressively increased the river discharge to 485 m^3^/s at the start and swiftly climbed to 808 m^3^/s when high tide arrived. With the chlorine level managed, the daily average discharge in this scheme was 493 m^3^/s.

The third tide scenario was an actual tide that took place in 2018. The observed chlorine level in the third tide scenario surpassed the threshold for nine days, which posed a major threat to the residential and production water safety (Figure 10b). The seawater intrusion suppression scheme boosted the river discharge to around 500 m^3^/s at the beginning, increased it to a maximum of 698 m^3^/s when high tide arrived, and then gradually decreased it to approximately 490 m^3^/s at the end, as shown in Figure 10c. Following high tide, this scheme kept the chlorine level at about 240 mg/L, and the daily average discharge of this scenario was 531 m^3^/s.

### 3.3. Practicable Seawater Suppression Scheme

Maintaining a consistent flow was better for the reservoir’s operation because it made scheduling easier [42]. Thus, the practicable river discharge for saltwater suppression was set to three phases, with the first phase being no less than 490 m^3^/s before the high tide arrival, the second phase being no less than 650 m^3^/s during the high tide, and the third phase being a return to no less than 490 m^3^/s after the high tide arrival. This scheme took into account the maximum and average river discharge in the first tide scenario, the average tidal range, and level for each month in five dry years, as seen in the previous section. The complete unit output flow range of the upstream power station of 600–700 m^3^/s made this concept compatible for power generation as well.

Since the influence of river discharge on chlorine level was time-dependent, it was necessary to determine the ideal moment to increase the river discharge. Assuming that, *N* days before the arrival of the high tide, the river discharge went from 490 m³/s to 650 m³/s for 6 days. The outcomes of the model ran are displayed in Figure 11. The best results for saltwater intrusion suppression occurred when *N* = 4, meaning that the river discharge rose four days before the arrival of high tide, and the maximum chlorine level was 248 mg/L. This outcome was more in line with the finding by Lu et al. according to whom the ideal time for seawater pressure was just before to the onset of high tide [43].

Based on the above analysis, as shown in Figure 12, this study has developed the seawater intrusion suppression scheme for the downstream of the Mingjiang River. Since the arrival date of the high tides can be predicted, we can regulate the river discharge with the upstream reservoirs according to the scheme.

There were 16 events of chlorine level exceeding the threshold caused by seawater intrusion in five dry years. Table 1 displayed the outcomes of simulations of the actual intrusions before and after the seawater intrusion suppression scheme was applied. The simulations, as can be observed, muted every intrusion event to varying degrees. A total of 75% of the total number of seawater intrusion events, or 12, had been eliminated, and the chlorine level had lowered to below the national threshold. This means that the frequency of seawater intrusion in the downstream of the Minjiang River would have significantly reduced if the scheme had been applied.

The scheme could successfully reduce a seawater intrusion risk when it had low intensity, modest peaks, and a brief duration. In the simulation, 62.5% of the chlorine levels were brought down to or below the very safe threshold of 150 mg/L. However, when the seawater intrusion was of high intensity, lengthy duration, with huge peaks, as in events 10 and 12–14 in Table 1, the chlorine level in the simulation failed to go below the threshold. However, the peak and duration indicators for chlorine level were substantially lower than the original readings in all four high-intensity occurrences, indicating that the risk posed by seawater intrusion could have been significantly decreased.

## 4. Conclusions

The daily average discharge, daily maximum tidal range, and daily minimum tidal level are the three critical influencing factors on the estuary chlorine level. From the Pearson correlation analysis, the effect of these factors on the chlorine level of the present day is cumulative, including the flow from the previous five days and the tide influence from the prior four days. Based on data from five dry years, a seawater intrusion suppression model was developed in this study. The model demonstrated good performance throughout both training and testing and was able to mimic chlorine level change, which was used to analyze seawater intrusion suppression schemes in different scenarios.

The chlorine level statistical model was used to calculate chlorine levels at different flows and tidal ranges. The result proved that the chlorine level tended to be low in situations where the flow was high and the tidal range was small. To keep the chlorine level unchanged when the tidal range started to grow, the flow had to be increased. When the tidal range was 4.0 m, the critical river discharge for chlorine level equal to 250 mg/L was 440 m^3^/s. When the tidal range reached 4.3 m, the critical river discharge was 520 m^3^/s and the growth rate of discharge was 26.7. The growth rate’s irregularity was important to note. When the tidal ranges were between 4.3–4.6 m and 4.6–4.9 m, the growth rates were 10.0 and 33.3, respectively.

This study explored three seawater suppression schemes with the minimum total runoff for three different tidal scenarios, taking into consideration the inadequate runoff in the downstream of the Minjiang River during the dry period. The minimum daily average discharge for the three cases was, respectively, 487 m^3^/s, 493 m^3^/s, and 531 m^3^/s.

Combining all of the above experimental findings, the practicable seawater suppression strategy for the Minjiang River estuary was established as follows to make it easier to regulate upstream reservoirs. The flow of the dispatch was at least 490 m^3^/s at first, and it climbed to at least 650 m^3^/s 4 days before the arrival of the high tide, which lasted for 6 days. Then, it decreased to 490 m^3^/s until the dispatch concluded. After simulation, the scheme was able to successfully reduce all seawater intrusion frequencies and intensities, and removed 75% of the measured seawater intrusion incidents.

## Figures and Tables

**Figure 1 ijerph-20-05211-f001:**
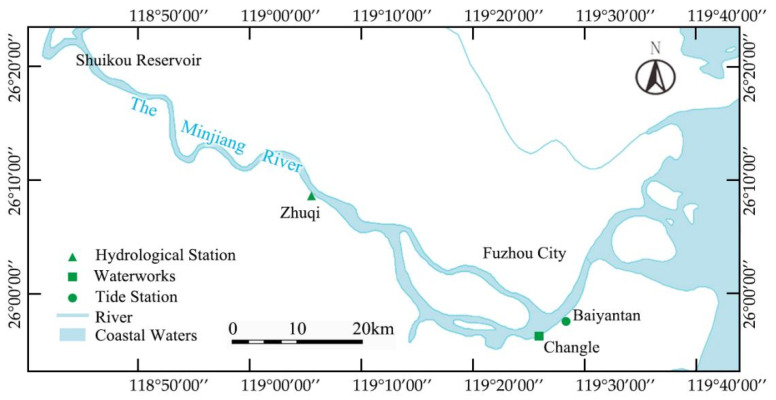
Overview of the study area.

**Figure 2 ijerph-20-05211-f002:**
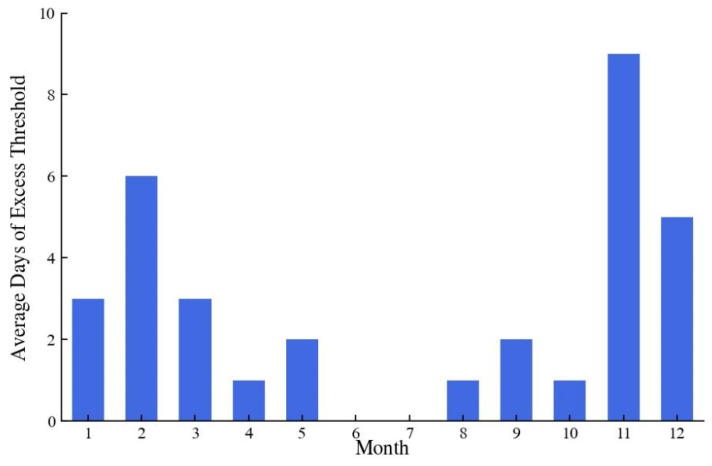
Average days of excess chlorine level for each month.

**Figure 3 ijerph-20-05211-f003:**
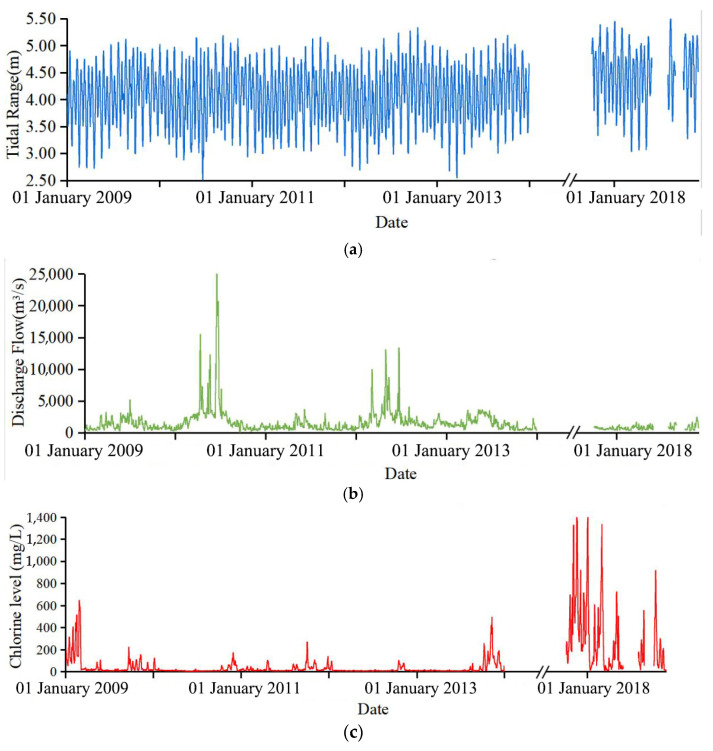
Time series of tide, flow, and chlorine level: (**a**) Tidal range; (**b**) Discharge flow; (**c**) Chlorine level.

**Figure 4 ijerph-20-05211-f004:**
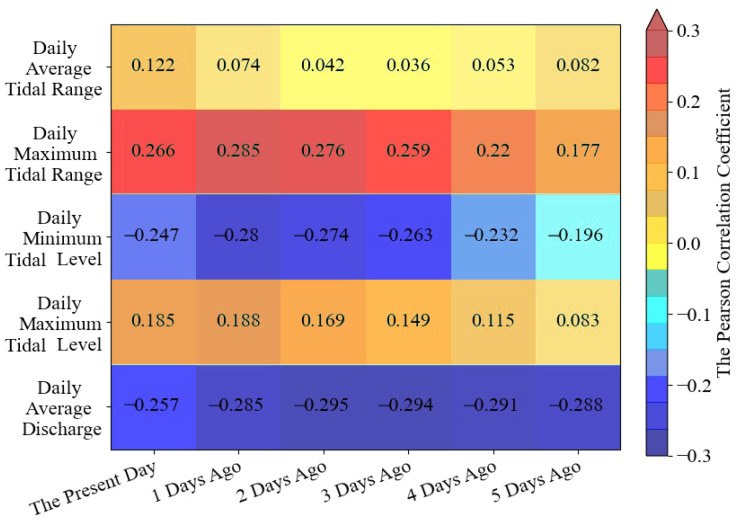
Pearson correlation coefficients between influence factors and chlorine level.

**Figure 5 ijerph-20-05211-f005:**
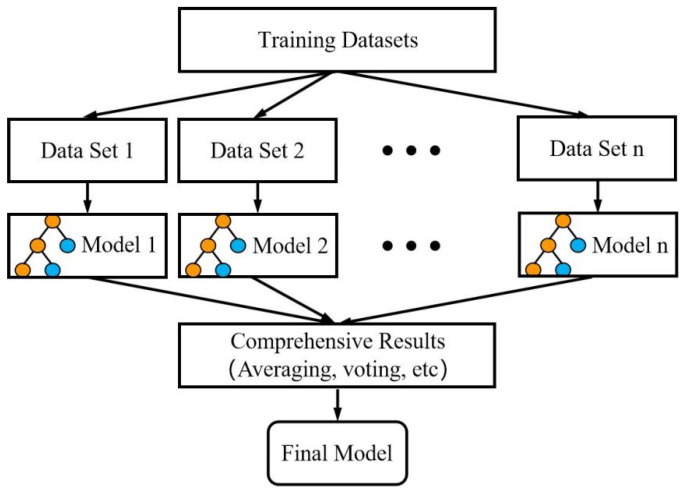
Basic structure of random forest.

**Figure 6 ijerph-20-05211-f006:**
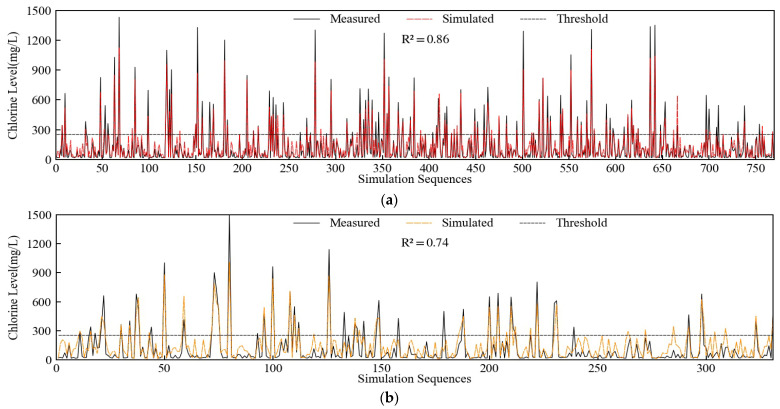
Simulation results: (**a**) Training datasets; (**b**) Test datasets.

**Figure 7 ijerph-20-05211-f007:**
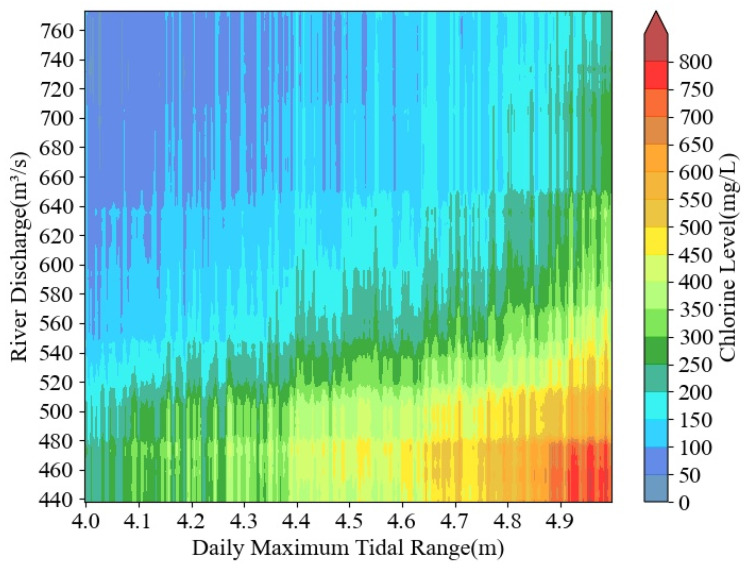
River discharge—tidal range—chlorine level relationship.

**Figure 8 ijerph-20-05211-f008:**
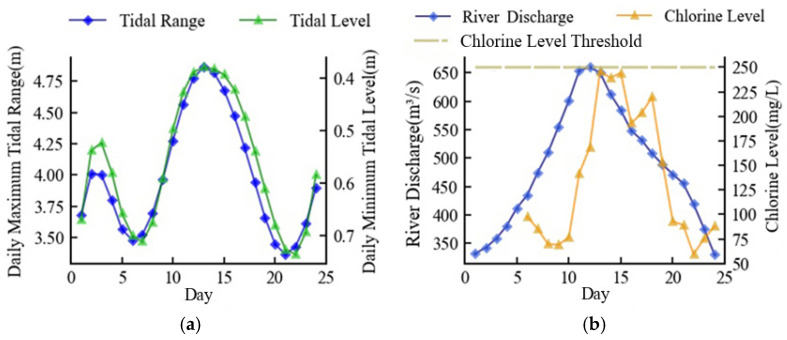
Minimum discharge scheme in the first tide scenario: (**a**) simulation of tide; (**b**) optimized river discharge and simulation of chlorine level.

**Figure 9 ijerph-20-05211-f009:**
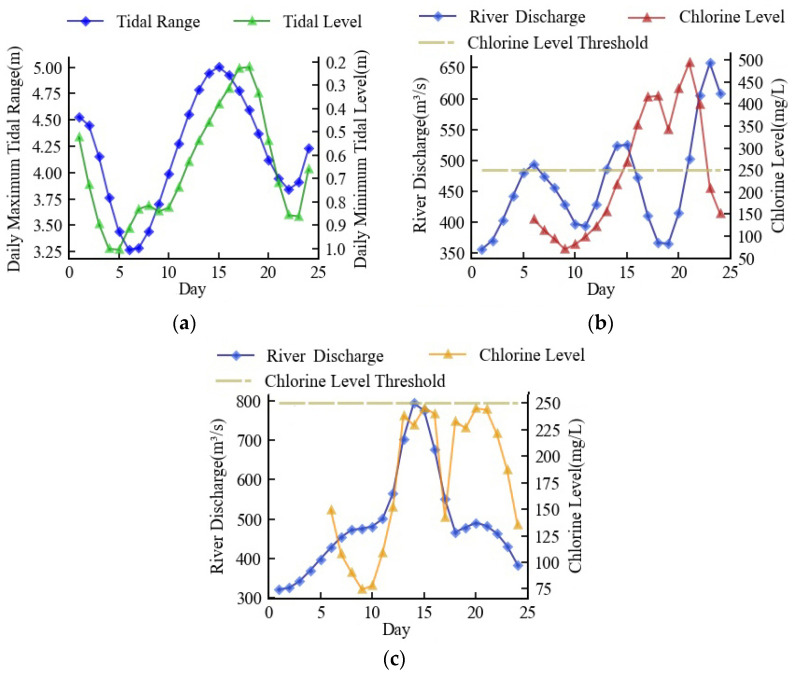
Minimum discharge scheme in the second tide scenario: (**a**) measured tide; (**b**) measured river discharge and chlorine level; (**c**) optimized river discharge and simulated chlorine level.

**Figure 10 ijerph-20-05211-f010:**
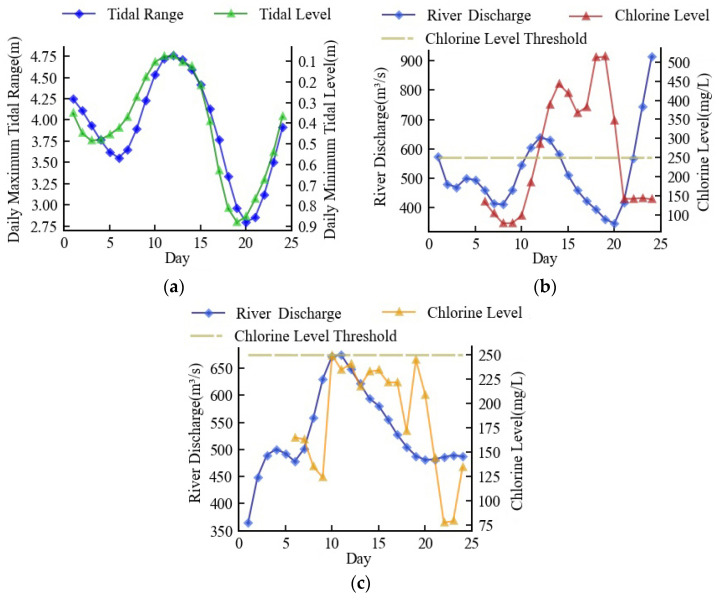
Minimum discharge scheme in the third tide scenario: (**a**) measured tide; (**b**) measured river discharge and chlorine level; (**c**) optimized river discharge and simulated chlorine level.

**Figure 11 ijerph-20-05211-f011:**
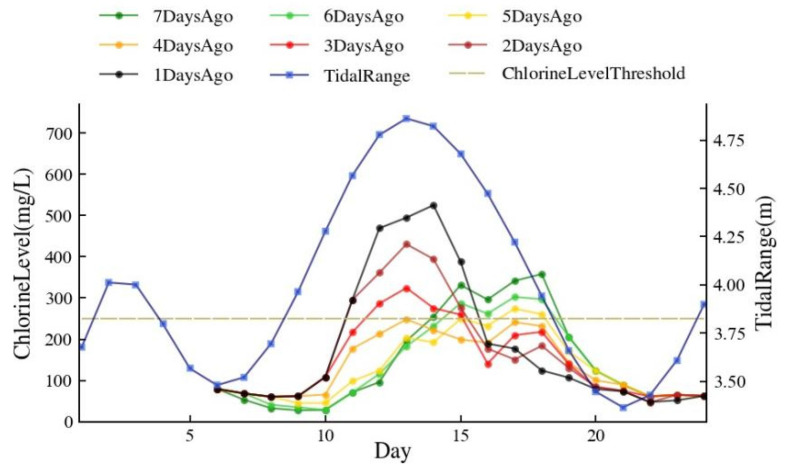
The variation of chlorine level when the river discharge is increased in advance of *N* days before the arrival of the high tide.

**Figure 12 ijerph-20-05211-f012:**
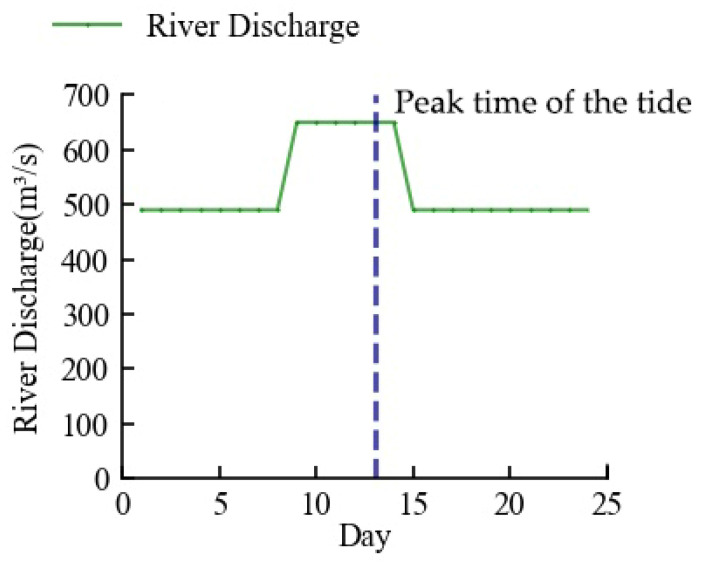
Practicable seawater suppression scheme.

**Table 1 ijerph-20-05211-t001:** Comparison of measured chlorine level and simulated chlorine level after application of seawater suppression scheme.

No.	Occurrence Time	Measured Chlorine Level		Simulated Chlorine Level	
		Max. (mg/L)	Exceeded Time (Day)	Max. (mg/L)	Exceeded Time (Day)
1	Early January 2009	312.13	3	67.77	0
2	Late January 2009	404.05	4	67.77	0
3	Early February 2009	514.21	9	149.43	0
4	Late February 2009	645.04	8	90.75	0
5	Late September 2011	267.55	2	90.75	0
6	Early October 2013	254.83	1	87.54	0
7	Late October 2013	493.37	8	63.48	0
8	Early November 2017	695.45	8	90.45	0
9	Late November 2017	1457.17	19	70.22	0
10	Early December 2017	917.13	11	495.52	5
11	Late December 2017	814.79	15	208.57	0
12	Early January 2018	1524.89	13	571.04	5
13	Early February 2018	605.67	4	426.88	3
14	Early March 2018	1332.31	13	431.94	5
15	Late April 2018	722.41	9	241.72	0
16	Early October 2018	913.91	10	149.89	0

## Data Availability

The data presented in this study are available upon request from the corresponding author.

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
