# Peer review of "Research on Seawater Intrusion Suppression Scheme of Minjiang River Estuary"

_ijerph, 2023, doi:10.3390/ijerph20065211_

Round 1

Reviewer 1 Report

This paper explored modeling seawater intrusion using machine learning. I think readers should be interested in this research. However, several aspects could be further improved in order to having it published in this journal. Therefore, my recommendation is major revision. The main questions I encountered when reading the manuscript are as follows:

Major concerns:

1、The Abstract is not well written. The authors do not explain the motivation of their study. Whats the problem of previous studies on seawater intrusion? Why use random forest algorithm?

2The Introduction is also not well-written. The authors do not explain why they use Random Forest Algorithm. Note that there are numerous powerful machine learning method, such as ANN, SVM. Does RFA perform better than ANN and SVM in the Minjing River estuary? If RFA does, please add the related results in the paper.

3Section 2, please add a picture to display time series of local river flow, water levels and chlorine levels.

4Introduction: ‘Those statistical models based on machine learning had advantages like quick calculation, high accuracy, and wide adaption ranges, which resulted in successful outcomes in a number of seawater intrusion studies with sparse data.’

I do not understand this sentence. In my understanding, machine learning methods need numerous data, if data is not ample, how can these methods obtain enough information from these data?

Minor concerns:

Language can be further improved. Tense should be unified and line number should be added.

1、Abstract: ...seawater intrusion risk and effectively reduced the chlorine level for the rest 25% events...  

Change reduced to reduce

2、Page1, introduction: ‘The issue of seawater intrusion have gradually spread across the globe in recent years due to increased human activity, the influence of sea level rise, and climate changed.’

Change have to has, change activity to activities, change climate changed to climate change

Because this paper does not have line number, thus, I only list part problematic sentences.

Author Response

Please check it in the attachment.

Reviewer 2 Report

The manuscript: “Research on Seawater Intrusion Suppression Scheme of Minjiang River Estuary”

The Text is written more like a technical report than a scientific study, and the authors should state why the applied method is new and interesting for the readers.

The authors present an example of a water management, based on a simplistic model and correlations coefficients.  

No information is given on the state of art in this field.

There are many errors in the English text which indicates that a native speaker of English should correct the text

Observations

Figure 1 is of low quality and all locations, mentioned in the text must be included in the figure.

Figure 3. Pearson correlation coefficients between influence factors and chlorine level.

What do the colors indicate? What is the Confidence Level?

Figure 5. Simulation results: (a) Training datasets; (b) Test datasets.

The size of the figure is too small, it is difficult to read.

The number of several citations is not included in the text.

Minor observations:

Introduction

This threatened the freshwater demand in coasts, especially in some water scarce areas.

Change into:

This threatened the freshwater demand on coasts, especially in some water-scarce areas.

The issue of seawater intrusion have gradually spread across the globe in recent years due to increased human activity, the influence of sea level rise, and climate changed [3].

Change into:

The issue of seawater intrusion has gradually spread across the globe in recent years due to increased human activity, the influence of sea level rise, and climate change [3].

Seawater intrusions were reported in coastal aquifers along both the east and west coasts of the United States [4,5]. Intrusions occurred in Mexico, Australia, and Libya as well [6-8].

Change into:

Seawater intrusions were reported in coastal aquifers along both the east and west coasts of the United States [4 ,5]. Intrusions occurred in Mexico, Australia, and Libya [6-8].

Also in several regions of Europe and Asia, seawater intrusion occurred and would be a threat for the future [9-12].

Change into:

Also in several regions of Europe and Asia, seawater intrusion occurred and would be a threat in the future [9-12].

Xu et al. assumed that different tides and runoff intensities would impact on the mixing type of seawater and freshwater, and built a MIKE21 model to simulated runoff, tide  and wind on the activity and concentration of the seawater [16].

Change into:

Xu et al. assumed that different tides and runoff intensities would impact the mixing type of seawater and freshwater, and built a MIKE21 model to simulate runoff, tide,  and wind on the activity and concentration of the seawater [16].

They then used this model to explore the best timing for increasing river discharge for suppress seawater intrusion [17]. Wang et al. considered the cumulative effect of the above factors on chlorine level and developed a more realistic chlorine level model in the Pearl River estuary [18].

Change into:

They then used this model to explore the best timing for increasing river discharge for suppressing seawater intrusion [17]. Wang et al. considered the cumulative effect of the above factors on chlorine levels and developed a more realistic chlorine level model in the Pearl River estuary [18].

 Meanwhile, the concept of critical river discharge for suppress seawater intrusion, the minimal river flow necessary to ensure that chlorine level in the mixed river water did not exceed the threshold, have been gradually put forth [22, 23].

Change into:

Meanwhile, the concept of critical river discharge for suppress seawater intrusion, the minimal river flow necessary to ensure that chlorine level in the mixed river water did not exceed the threshold, has been gradually put forth [22, 23].

These studies focused mainly on the suppression effect of flow on tides, but could not give solutions under the water scarcity situation.

Change into:

These studies focused mainly on the suppression effect of flow on tides, but could not give solutions to the water scarcity situation.

The statistical models, which were mostly used for the study of the seawater suppression strategy or the prediction of seawater intrusion, had been based on actual data, taking into account statistical correlation among variables[29].

Change into:

The statistical models, which were mostly used for the study of the seawater suppression strategy or the prediction of seawater intrusion, had been based on actual data, taking into account the statistical correlation among variables   [29].

Numerical modeling required large amount of basic data, which was difficult to use in areas where data were lacking.

Change to:

Numerical modeling required a large amount of basic data, which was difficult to use in areas where data were lacking.

With the rise of computer technology, more and more studies began to utilize statistical models based on machine learning, of which the most popular methods included artificial neural network, support vector machin, random forest algorithm, etc[30-32].

Change into:

With the rise of computer technology, more and more studies began to utilize statistical models based on machine learning, of which the most popular methods included artificial neural networks, support vector machines, random forest algorithms, etc. [30-32].

The amount of sand from upstream has been reduced, river scouring has gotten worse, and riverbed undercutting has become more serious in recent years asa results of the building of upstream reservoirs and downstream sand mining activities [36].

Change into:

The amount of sand from upstream has been reduced, river scouring has gotten worse, and riverbed undercutting has become more serious in recent years as a results of the building of upstream reservoirs and downstream sand mining activities [36].

Consequently, the estuary's original hydrodynamic properties have been changed, the tide affected area constantly moved upstream, and the chlorine level frequently exceeded the threshold of norms, posing major threats to the local water supply. In this study, a statistical model of…….

Change into:

Consequently, the estuary's original hydrodynamic properties have been changed, the tide-affected area constantly moved upstream, and the chlorine level frequently exceeded the threshold of norms, posing major threats to the local water supply. In this study, a statistical model of…...

The upstream Shuikou reservoir, which was built in 1993 with a catchment area of 52,438 km2, controls the downstream flow and is used for power generation, AND flood control  and irrigation.

Change into:

The upstream Shuikou reservoir, which was built in 1993 with a catchment area of 52,438 km2, controls the downstream flow and is used for power generation, and flood control,  and irrigation.

The annual flow of Minjiang River can reach a maximum of 1733 m3/s, while the annual discharge is around 55 billion m3.

Change into:

The annual flow of the Minjiang River can reach a maximum of 1733 m3/s, while the annual discharge is around 55 billion m3.

The salty tide frequently affect the waterworks in downstream of Minjiang River in dry years,  and causing chlorine level exceeded the threshold

Change to:

The salty tide frequently affects the waterworks in the downstream of Minjiang River in dry years and causing chlorine level exceeded the threshold

The number of days that have exceeded the threshold have progressively grown in recent years.

Change into:

The number of days that have exceeded the threshold has progressively grown in recent years.

Fluctuations in river discharge and tide in the downstream of Minjiang River are the main factors affecting seawater intrusion, which are used to seek the change of chlorine level in this study.

Change into:

Fluctuations in river discharge and tide downstream of Minjiang River are the main factors affecting seawater intrusion, which are used to seek the change of chlorine level in this study.

This plan can be supported by the conclusions of Chen et al. [J1] by finding that the water flow, which suppressed saltwater, constraint the tide's gradient force, which promoted infiltration of seawater [37].

Change into:

This plan can be supported by the conclusions of Chen et al. [J2] by finding that the water flow, which suppressed saltwater, constrains the tide's gradient force, which promoted the infiltration of seawater [37].

In the same approach, the model's tidal level data have chosen to be the minimum tidal level. Although the minimum tide level had a negative correlation value…

Change into

In the same approach, the model's tidal level data have been chosen to be the minimum tidal level. Although the minimum tide level had a negative correlation value…

As shown in Figure 3, their correlation coefficients were significantly lower than the other days. Therefore, to further improve the model's operational efficiency, data on tidal range and level that occurred from five days ago have been eliminated in this model. 

Change into

As shown in Figure 3, their correlation coefficients were significantly lower than on the other days. Therefore, to further improve the model's operational efficiency, data on the tidal range and level that occurred five days ago have been eliminated in this model. 

The decision coefficient R2, which have been established as follows and used to evaluate the precision of the model fit.

Change into:

The decision coefficient R2, which has been established as follows and used to evaluate the precision of the model fit.

And so on……

Author Response

Please check it in the attachment.

Round 2

Reviewer 1 Report

The paper can be accepted. But there are still some minor language probelms.

Line 10 change 'have' to 'has'

Line 11 change 'focused mainly on' to 'mainly focused on'

Also, in the response, the answer should be as detailed as possible.

Author Response

请参阅附件。

Reviewer 2 Report

Response to the authors

Point 1: The Text is written more like a technical report than a scientific study, and the authors should state why the applied method is new and interesting for the readers.

Response 1: We have provided a explanation in the abstract about why it is important to study ways to resist seawater intrusion in the Minjiang River Estuary. This is a real problem that urgently needs to be solved, and our peers have focused too much on the research mechanism without proposing a solution. In the introduction, we explained why we chose to use the random forest algorithm. Compared with other algorithms, the random forest algorithm has lower requirements for data sample size and good high-dimensional data processing capabilities, which meets the needs of Minjiang estuary research. Therefore, the random forest algorithm may not be the optimal algorithm, but it is the best algorithm that meets this research. (See lines 10-18, 84-94 and 103-109).

I do not doubt that the text has helped to find a solution for a “real” problem, but is it significant for the scientific community? I think the editor should decide.

Point 2: The authors present an example of a water management, based on a simplistic model and correlations coefficients. No information is given on the state of art in this field.

Response 2: There are many applications of machine learning in seawater intrusion research. We would show you some examples of machine learning has been widely used for numerical simulation, sensitivity analysis, pattern classification, prediction and other studies of seawater intrusion by citing literature. (See lines 76-83).

OK

Point 3: There are many errors in the English text which indicates that a native speaker of English should correct the text.

Response 3: We have corrected the errors in the English text that you pointed out and tried to make more revisions to the article text.

There are still errors to be corrected, see the following examples:

Line 83 ff

 … machine learning has been employed for multivariate analysis of nutrient distribution in water environment across temporal and spatial scales in recent decades,  and the amount of such studies has grown exponentially in recent years [30].

Should be:

…  machine learning has been employed for multivariate analysis of nutrient distribution in water environments across temporal and spatial scales in recent decades,  and the amount of such studies has grown exponentially in recent years [30].

Line 98 ff

Moreover, because it could handle collinearity between variables, random forest algorithm did not overfitting when dealing with high-dimensional data …

Should be:

Moreover, because it could handle collinearity between variables, the random forest algorithm did not overfit when dealing with high-dimensional data …  

Line 111 - 113

Therefore, studying a scheme to suppress seawater intrusion and mitigate its impact on municipal water supply in Fuzhou is of practical significance.

Should be:

Therefore, studying a scheme to suppress seawater intrusion and mitigate its impact on the municipal water supply in Fuzhou is of practical significance.

Point 4: Figure 1 is of low quality and all locations, mentioned in the text must be included in the figure.

Response 4: We have replaced Figure 1 and hope you can read it clearly. And the locations mentioned in the text are all on the map.

OK

Point 5: Figure 3. Pearson correlation coefficients between influence factors and chlorine level. What do the colors indicate? What is the Confidence Level?

Response 5: We have added a color bar to Figure 3 to explain the meaning of different colors. We also added a paragraph about the correlation coefficient in the text, so that readers can better understand the level of each element.(See lines 158-160).

Correlation Confidence Interval of the Pearson correlation coefficient depends on the number of samples and should be calculated. This information should be included in the text.

Point 6: Figure 5. Simulation results: (a) Training datasets; (b) Test datasets. The size of the figure is too small, it is difficult to read.

Response 6: We have changed the line type of the simulated data in Figure 5 to make it more clearly show our model’s fitting ability.

The figure is better to read now, but I recommend increasing the size.

Point 7: The number of several citations is not included in the text.

Response 7: We included the number of several citations.

OK
